# Radiogenomics: Contemporary Applications in the Management of Rectal Cancer

**DOI:** 10.3390/cancers15245816

**Published:** 2023-12-12

**Authors:** Niall J. O’Sullivan, Hugo C. Temperley, Michelle T. Horan, Alison Corr, Brian J. Mehigan, John O. Larkin, Paul H. McCormick, Dara O. Kavanagh, James F. M. Meaney, Michael E. Kelly

**Affiliations:** 1Department of Radiology, St. James’s Hospital, D08 NHY1 Dublin, Ireland; horanmi@tcd.ie (M.T.H.);; 2School of Medicine, Trinity College Dublin, D02 PN40 Dublin, Ireland; 3The National Centre for Advanced Medical Imaging (CAMI), St. James’s Hospital, D08 NHY1 Dublin, Ireland; 4Department of Surgery, St. James’s Hospital, D08 NHY1 Dublin, Ireland; temperlh@tcd.ie; 5Department of Surgery, Tallaght University Hospital, D24 NR0A Dublin, Ireland; 6Department of Surgery, Royal College of Surgeons, D02 YN77 Dublin, Ireland; 7Trinity St. James’s Cancer Institute (TSJCI), D08 NHY1 Dublin, Ireland

**Keywords:** radiomics, radiogenomics, oncology, rectal cancer, survival, recurrence, treatment response

## Abstract

**Simple Summary:**

Rectal tumour biological characteristics play an important role in determining treatment regimen, predicting treatment response and predicting prognosis. Currently, obtaining tumour biological information requires costly, invasive and time-consuming genetic testing. Radiogenomics, referring to the extraction of imaging biomarkers that may serve as identifiers for specific biological characteristics, serves as a non-invasive alternative to genetic testing. Our study aims to collate the current evidence for radiogenomics in the field of rectal cancer, highlighting strengths and weaknesses of individual studies.

**Abstract:**

Radiogenomics, a sub-domain of radiomics, refers to the prediction of underlying tumour biology using non-invasive imaging markers. This novel technology intends to reduce the high costs, workload and invasiveness associated with traditional genetic testing via the development of ‘imaging biomarkers’ that have the potential to serve as an alternative ‘liquid-biopsy’ in the determination of tumour biological characteristics. Radiogenomics also harnesses the potential to unlock aspects of tumour biology which are not possible to assess by conventional biopsy-based methods, such as full tumour burden, intra-/inter-lesion heterogeneity and the possibility of providing the information of tumour biology longitudinally. Several studies have shown the feasibility of developing a radiogenomic-based signature to predict treatment outcomes and tumour characteristics; however, many lack prospective, external validation. We performed a systematic review of the current literature surrounding the use of radiogenomics in rectal cancer to predict underlying tumour biology.

## 1. Introduction

The gold-standard treatment for locally advanced, non-metastatic rectal cancer includes neoadjuvant chemoradiotherapy (NACRT), total mesorectal excision (TME) and adjuvant chemotherapy (AC) [1,2]. The primary goal of treatment is to achieve local disease control, reduce tumour volume and minimise the risk of distant metastases [3]. While this multimodal treatment approach has offered improvements in local control and sphincter preservation, it has had little effect on distant recurrence and overall survival [4,5]. Despite significant advancements in the management of LARC, several challenges and limitations persist [6]. The optimum chemotherapeutic treatment regimen for patients with LARC remains a debate, with many clinical factors playing a role in final treatment decision [7]. A personalised therapeutic approach, with the aid of radiogenomics, carries the potential to revolutionise the way we treat this cohort of patients indefinitely [8]. Predicting tumour gene expression from pre-therapeutic imaging could alleviate the need for invasive genetic testing, ultimately giving us the ability to predict treatment response and adjusting regimens accordingly [9].

Radiogenomics, a sub-domain of radiomics, refers to the prediction of underlying tumour biology using non-invasive imaging markers [10]. A radiogenomic workflow involves several key steps, ultimately involving the conversion of radiological images into high-dimensional mineable data in a high-throughput fashion [11,12]. Following acquisition of images, raw imaging data must be pre-processed via manual, semi-automated or fully automated machine learning methods to facilitate segmentation of the regions of interest (ROI) [13,14,15]. Radiomic features can then be extracted from the ROIs using various feature extraction software such as PyRadiomics v3, Computational Environment for Radiological Research (CERR) and Image Biomarker Explorer (IBEX) [11,16,17,18]. Radiomic features can be broadly subdivided into textural, morphological and functional radiomics [12]. Texture analysis refers to the measure of variation of pixel intensity throughout a given image or ROI [19]. Morphological features include spherical disproportion, maximum 3D diameter and surface volume ratio [20]. Functional radiomics refers to features extracted from functional imaging, demonstrating regional, time-varying changes within tumour tissue [21]. Extracted features are then subjected to rigorous statistical analysis in order to identify those highly correlated with the expected outcome [22]. Finally, machine learning classifiers and statistical methods are then implemented to ultimately develop predictive and prognostic radiogenomic models [11]. Model performance is generally evaluated using the receiver-operating characteristic (ROC) and area under curve (AUC) [23]. 

Radiogenomics may facilitate a reduction in the high costs, workload and invasiveness associated with traditional genetic testing, via the development of ‘imaging biomarkers’, potentially serving as an alternative ‘liquid-biopsy’ in the determination of tumour biological characteristics and oncogenes status [24]. Radiogenomics also harnesses the potential to unlock aspects of tumour biology which are not possible to assess by conventional biopsy-based methods, such as full tumour burden, intra-/inter-lesion heterogeneity and the possibility of providing the information of tumour biology longitudinally [11,25]. The role of this novel technology has been investigated in several areas of oncology, including glioblastoma, breast cancer, renal cell carcinoma and colorectal cancer [12,26,27]. While numerable studies have demonstrated the feasibility and efficacy of radiogenomic model construction, many lack prospective validation in an external setting [28,29]. The aim of our study is to present the current evidence surrounding the use of radiogenomics in rectal cancer, highlighting strengths, weaknesses and heterogeneity of radiogenomic workflow between individual studies in order to guide future research within this novel field of diagnostic medicine.

## 2. Methods

### 2.1. Study Design and Reporting Guidelines

This study is a systematic review of randomised and non-randomised trials and follows the Preferred Reporting Items for Systematic Reviews and Meta-Analyses (PRISMA) reporting guidelines. PRISMA 2020 is a 27-item checklist aiming to provide structure to systematic reviewers, facilitating transparency in the research question, methodology and results [30].

### 2.2. Search Strategy

The following databases were searched as part of the systematic review in March 2023: Medline, EMBASE and Web of Science. Terms included (radiomic* OR radiogenomic*) AND (rect*) AND (oncolog* OR treatment OR surviv*). The last date of search was 20 September 2023. The grey literature was also searched to further identify other suitable publications.

### 2.3. Eligibility Criteria

Studies were assessed for eligibility based on the following inclusion criteria. Randomised controlled trials (RCTs) or cohort studies investigating the use of radiogenomics to predict biological tumour characteristics in patients with rectal cancer were included in our analysis. Biological tumour characteristics were defined as gene mutations, tumour grade and tumour differentiation. Case reports, case series and conference abstracts were excluded.

### 2.4. Study Selection, Data Extraction and Critical Appraisal

A database was established using EndNote X9^TM^ reference management software. A comprehensive assessment of search outputs was performed by NOS and HCT independently of each other. Following elimination of duplicate studies, titles were screened and assessed for relevance as per our eligibility criteria. Abstracts were subsequently scrutinised for potential relevance, based on our pre-defined inclusion and exclusion criteria. The full texts of eligible abstracts were then subjected to further analysis, utilising the same set of criteria. Any disagreements between the two reviewers, NOS and HCT, were resolved via open discussion, with the final decision being made by the senior author, MK. Rejected studies were categorised in our database according to reason for exclusion. 

For efficient data extraction and storage, Covidence (Cochrane Collaboration) was used for screening and data extraction [31]. Data collection was independently conducted by two reviewers (NOS and HCT). Data was recorded under specific headings, including study details, study design, population, intervention, comparison groups and outcomes. Any discrepancies between the reviewers were addressed through open discussions and resolved with the guidance of the senior author, MK.

A critical appraisal of the methodological quality and risk of bias was performed on the included studies by three reviewers (NOS, HCT and MH) independently, using the Quality Assessment of Diagnostic Accuracy Studies 2 (QUADAS-2) and Radiomics Quality Score (RQS) [23,32]. Any discrepancies were resolved by open discussion, with the final decision being made by the senior author, MK. The RQS tool assesses the quality of radiomics-based studies on 16 criteria, with a maximum score of 36 points. Studies with a higher amount of points are deemed of superior quality than those lower on the scale. The QUADAS-2 score assesses risk of bias within studies based on four parameters: patient selection, index test, reference standard and flow and timing. Studies are assigned low, medium, high or unclear risk to each of these criteria. This scoring system also addresses applicability concerns of developed signatures.

### 2.5. Systematic Review Registration

Our systematic review was registered on PROSPERO, an international database of prospectively registered systematic reviews in health-related research, in March 2023 (CRD42023412743). PROSPERO provides a comprehensive collection of systematic review protocols in an attempt to avoid duplication of effort, reduce reporting bias and promote overall transparency of research [33].

## 3. Results

### 3.1. Search Results

The literature search described above yielded a total of 673 results (Appendix A). Following the removal of 203 duplicates, 470 studies were screened. After the initial screen, 86 abstracts were reviewed and assessed for eligibility, of which 30 were selected for full text review. From these 30 full texts, a total of 13 studies met the inclusion criteria and were included in our qualitative analysis. Of the seventeen full-texts excluded, nine were for an incorrect study design (six conference abstracts, three case series), four were of an incorrect population (four colon cancer), three failed to develop radiogenomic-based signatures and one study had no measurable outcomes. Due to considerable heterogeneity between study outcomes, a quantitative analysis was not performed, limiting the generalisability of findings. Therefore, the results should be interpreted with caution.

### 3.2. Methodological Characteristics and Quality of Studies

Eleven out of the thirteen included studies were retrospective in nature [34,35,36,37,38,39,40,41,42,43,44]. One study included data collected both retrospectively and prospectively [29]. The remaining study was performed entirely prospectively [28]. Table 1 summarises the methodological characteristics of the included studies. Data quality, assessed using the RQS and QUADAS-2 tools, was generally satisfactory. All studies were deemed low risk of bias based off the QUADAS-2 tool, while 69% (n = 9/13) of the included studies received an RQS score >30%. A detailed explanation of the tools and breakdown of the results can be found in Appendix A (Appendix A).

### 3.3. Participant Characteristics

The total number of participants from the thirteen included studies was 2378. Nine studies included both training and validation sets within their studies, whereas the remaining four studies included only a training cohort. Overall, 1611 patients constituted the training sets and 767 the validation sets across included studies. Out of 2378 participants, 1487 were male and 891 were female. Basic participant characteristics are outlined in Table 2.

### 3.4. Acquisition Parameters

Multi-parametric MRI (mpMRI) was the most commonly used modality, implemented in seven studies overall. The remaining studies developed their models from T2-weighted MRI or contrast-enhanced CT. Field strength ranged from 1.5 to 3.0 Tesla (T) across included studies. Median slice thickness was 3.5 mm (1.25 mm–5 mm). Contrast administration was reported in six studies. The agent of choice was gadolinium in MRI studies and Ultravist (iopromide) and IsoVue (iopamidol) in contrast-enhanced CT studies. Full acquisition parameters are illustrated in Table 3.

### 3.5. Development of Signatures

Exact feature extraction methods varied across studies; however, a relatively similar pathway was followed across the board. Regions of interest (ROI) were first segmented manually by experienced radiologists in all included studies. ITK-SNAP and 3D-Slicer were the most commonly utilised segmentation software across the included studies, employed in five and two studies, respectively. Radiomic features were then extracted from regions of interest using various radiomics software. MATLAB and PyRadiomics were the most frequently utilised software for feature extraction, accounting for five and two studies, respectively. Specific software used in individual studies for image segmentation and feature extraction is demonstrated in Table 4. Reliability of radiomics features was assessed in several studies by intraclass correlation coefficient (ICC) [28,34,36,38,39,43]. Acceptable ICC values ranged from 0.75–0.8. Features with an unsatisfactory ICC were discarded. A Mann–Whitney U (MWU) test was performed in several studies on remaining features. Features with a *p*-value < 0.05 were preserved. Finally, the least absolute shrinkage and selection operator (Lasso) logistic regression model was subsequently applied to features to remove sub-optimal features. Remaining features were used to construct individual radiogenomic signatures.

### 3.6. Performance of Signatures

Performance of models, estimated using the receiver operating characteristic (ROC) curve and summarised as the area under curve (AUC), ranged between studies. Median AUC, sensitivity and specificity (validation cohort) across the included studies were 0.723 (0.651–0.895), 0.762 (0.438–0.941) and 0.643 (0.422–1), respectively. Table 4 llustrates the performance of each model in predicting primary outcome. All included studies had at least satisfactory performance in predicting outcomes. Higher AUC values correlate to better model performance in predicting primary outcome.

## 4. Discussion

Our review demonstrates the feasibility and efficacy of developing radiogenomic-based signatures to predict biological tumour characteristics in patients with rectal cancer from raw, pre-treatment radiological images. All signatures developed within included studies predicted their primary outcome with at least modest accuracy (AUC range 0.651–0.895). While these are promising results, significant heterogeneity in population, workflow and primary outcome between studies makes direct comparison and further replication difficult. Standardisation in the form of open-source scans, segmentations and code will allow for controlled external validation in alternative institutions, paving the way for eventual application within the clinical setting [23,45].

While studies investigating the use of radiogenomics in rectal cancer remain scarce, those demonstrating the use of radiomics to predict response to neoadjuvant chemotherapy and overall survival in rectal cancer are emerging [46]. In a recent systematic review by Staal et al., a total of 76 studies investigating the development of radiomics-based signatures to predict treatment and oncological outcomes were included in their analysis [46]. Radiomic-based signatures were developed using MRI, CT and FEG-PET/CT in 41, 30 and 10 studies, respectively. Their review demonstrated that MRI-based radiomics studies tended to be of higher quality when assessed using the RQS and QUADAS tools. This review highlighted the exponential growth in both number of radiomics studies in the last decade, as well as the complexity of these studies in terms of specific features extracted. Methodological quality was generally poor across their included studies, with 77% of studies demonstrating an RQS score < 30%. This is in contrast to our review, where only 30% of studies scored < 30%; however, several of these were borderline. Low scores are generally attributed to lack of external validation, prospective setting and feature reduction [46]. Studies which perform well on the RQS questionnaire do so when sufficient feature reduction and validation is performed. The authors concluded that radiomic signatures had overall adequate performance in the prediction of treatment outcome and survival in patients with colorectal cancer; however, future studies should focus on independent, external validation of existing signatures, as opposed to development of new models.

Treatment for advanced colorectal cancer can often be limited by complex genomic profiles that facilitate resistance to both targeted systemic agents and targeted monotherapies [47,48,49]. In the era of precision medicine, the efficacy of anti-EGFR agents in RAS wild-type patients and anti-PD-1 immunotherapies in patients with MMR deficiency or high MSI are well established [50,51,52,53]. Similarly, the resistance of tumours with KRAS mutations to anti-EGFR therapy are well documented, rendering tumours insensitive to this chemotherapeutic agent in many cases [54]. To date, genetic testing via next-generation sequencing or single-gene testing remains costly [55]. Once fully validated clinically, radiogenomics may be used to offset these costs in the future, by developing imaging surrogates for specific genetic signatures capable of predicting response to therapy and oncological outcomes [56,57]. Cost comparison between these techniques is yet to be investigated. 

Feature reliability remains one of the greatest obstacles to radiomics-based research [45]. Heterogeneity and uncertainty may arise from many areas within the complex radiomics workflow, ultimately hindering feature reproducibility, stability and validity [58,59,60,61,62,63]. In their systematic review of 481 studies investigating the utility of radiomics in medicine, Xue et al. attempted to define reliability via intraclass correlation coefficient (ICC) expression [45]. ICC is a measure of reliability commonly used in medical literature, which may be applied to radiomic features that have continuous values [45,64]. The authors conclude by offering several suggestions for researchers carrying out radiomics-based research in an effort to mitigate the pitfalls identified in their in-depth analysis of 481 manuscripts. Similarly, Koo et al. also provided clinical researchers with a practical step-by-step guideline to select the correct form of ICC in an effort to avoid selecting inappropriate ICC forms, which have the potential to result in misleading interpretations [64].

Overcoming variability in feature extraction and lack of reproducibility remains the primary barrier to success of future radiomics-based research [65]. In order to traverse these obstacles, future studies must focus on standardising imaging protocol (including contrast dose administration), having consistent acquisition parameters and using reconstruction kernels with lower noise levels [66]. Several studies have analysed radiomics features to determine which feature subtype tends to be the most reproducible in order to guide future radiomics-based research [67,68]. In their review of 41 articles, Traverso et al. assessed the repeatability and reproducibility of radiomics features derived from studies investigating the predictive utility of radiomics in non-small-cell lung cancer and oropharyngeal cancer. The authors concluded that first-order features, entropy in particular, were generally considered more reproducible than shape metrics or textural features [68]. Entropy refers to a logarithmic function of the ROI and has shown promise as a surrogate for tumour heterogeneity with good performance as a quantitative imaging biomarker for characterising cancer phenotype [69]. Similarly, Berenguer et al. found that out of 177 radiomics features extracted from studies performed on five CT scanners, 71 were considered reproducible. Despite this, only ten radiomics features were relevant and, therefore, retained [67].

Locally advanced rectal cancer remains a complex and molecularly heterogenous disease, often highly variable in its clinical and pathological landscapes [70]. To the best of our knowledge, at the time of writing this manuscript, no open-source fully automated machine learning algorithms have been developed and made available to researchers to aid in image segmentation. Heterogeneity and complexity in presentation of rectal tumours, particularly LARC, may have a knock-on effect on the efficacy and applicability of radiomics within this complex, evolving field of medicine. Despite having relatively similar cohorts and radiogenomic workflows, studies included in our analysis had highly variable results, potentially contributed to by the highly variable tumour and subsequent ROI and presentation. Large studies with vast amounts of imaging are required to build both machine learning algorithms as well as robust and reliable radiomics signatures in order to accurate predict tumour biology in this cohort of patients [71].

Our review is limited by the relatively small number of studies included in our analysis. Unfortunately, given the novelty of this field of medicine, this is unavoidable. Heterogeneity in radiogenomic workflow, such as variation in standardisation, acquisition parameters and segmentation programmes between institutions and primary outcome prevented us from performing a meta-analysis of included studies. Despite improved methodological quality compared to previous systematic reviews on radiomics-based research, overall RQS quality remained relatively low. Similarly, eleven of the included studies were retrospective in nature, potentially introducing further bias. All studies were included, regardless of quality, to offer an insight into all the existing literature on this novel topic, highlighting both strengths and weaknesses.

Ultimately, radiogenomics wields the potential to serve as an additional tool in the overall patient-counselling experience [72]. Rectal cancer treatment can profoundly affect a patient’s quality of life, often inducing significant morbidity [73]. Guiding patients through treatment options, particularly on a personalised level, may assist in the justification for certain aggressive treatment options that are likely to inflict morbidity on patients. Utilising these tools such as radiogenomic prediction models during the patient counselling experience gives patients an increased insight into their condition and helps them to manage treatment expectations. Radiogenomic prediction of tumour biological characteristics carries significant potential; however, several obstacles must be overcome prior to its eventual implementation into clinical practice. While the current progress indicates the feasibility and efficacy of developing radiogenomic-based models capable of predicting tumour biological characteristics, more large-scale prospective and ideally comparative studies are required before this technology and be validated and implemented into clinical practice.

## 5. Conclusions

Our review provides an in-depth analysis of the current literature investigating the use of radiogenomics to develop image surrogates for genetic aberrations in rectal cancer. Although workflow criteria and internationally set guidelines for statistical methodology need to be clarified, radiogenomics wields the potential to serve as an alternative, non-invasive approach to conventional genetic testing once clinically validated, in order to tailor therapy and facilitate personalised care in patients with advanced rectal cancer.

## Figures and Tables

**Table 1 cancers-15-05816-t001:** Methodological characteristics of the included studies.

Study	Country	Journal	Primary Outcome
Chen 2020 [34]	China	Journal of Magnetic Resonance Imaging	TOPO-IIα expression
Chen 2022 [28]	China	Abdominal Radiology	Aquaporin-1 expression
Horvat 2019 [35]	USA	European Journal of Radiology	APC, RASA1, ATM, BRCA2
Huang 2018 [36]	China	Academic Radiology	Tumour grade
Jeon 2021 [37]	Korea	Radiotherapy and Oncology	CD8+ TIL density
Jing 2022 [38]	China	BioMed Research International	MMR
Li 2022 [39]	China	Frontiers in Oncology	LRP-1 and survivin expression
Meng 2019 [40]	China	European Radiology	Tumour differentiation, Ki-67, HER-2, lymph node metastases and KRAS-2
Negreros-Osuna 2020 [41]	USA	Radiology: Imaging Cancer	BRAF
Oh 2020 [29]	Korea	Cancer Research and Treatment	KRAS
Zhang, G. 2021 [42]	China	Frontiers in Oncology	KRAS, NRAS, BRAF
Zhang, W. 2021 [43]	China	Annals of Translational Medicine	Microsatellite instability
Zhang, Z. 2021 [44]	China	Frontiers in Oncology	KRAS

**Table 2 cancers-15-05816-t002:** Participant characteristics.

Study	# Patients		Age (Mean ± Standard Deviation) [Years]		Male:Female
	T	V	T	V	
Chen 2020 [34]	85	37	58.6 ± 9.5	59.5 ± 14	76:46
Chen 2022 [28]	87	23	60.7 ± 12.5		65:45
Horvat 2019 [35]	65	-	57 ± 13.8		38:27
Huang 2018 [36]	222	144	61 ± 12.8		219:147
Jeon 2021 [37]	75	38	61 ± 9.5	64 ±12.5	78:35
Jing 2022 [38]	111	65	58.1 ± 11		115:61
Li 2022 [39]	70	30	68.3 ± 10.2	68.7 ± 9.6	64:36
Meng 2019 [40]	197	148	59.1 ± 12.2	61 ± 12.4	213:132
Negreros-Osuna 2020 [41]	145	-	61 ± 14		77:68
Oh 2020 [29]	60	-	61 ± 9.9		34:26
Zhang, G. 2021 [42]	108	94	59.9 ± 11.8		139:63
Zhang, W. 2021 [43]	327	164	MSS 60.5 ± 4.9 MSI 59.2 ± 12.8	MSS 60.7 ± 11.3 MSI 55.4 ± 12.1	318:173
Zhang, Z. 2021 [44]	59	24	55 ± 9.7		51:32

T: training set. V: validation set. MSS: microsatellite stability. MSI: microsatellite instability.

**Table 3 cancers-15-05816-t003:** Scanning parameters.

Study	Modality	Model	SLT	TR/TE (ms)	FOV (cm)	ACQ Matrix	ETL	Flip Angle (Degrees)	Contrast
Chen 2020 [34]	T2WI MRI	3.0T (Achieva, Philips or Netherlands)	4 mm	3000/90	19 × 19	272 × 228	19	-	-
Chen 2022 [28]	mpMRI	Siemens Prisma 3.0T	4 mm	4100/93	16 × 16	-	-	150	Gd 3 mL/s, 0.2 mmol/kg
Horvat 2019 [35]	mpMRI	GE Healthcare 1.5 or 3.0T	3 mm	4000–6000/120	18 × 18	-	-	-	-
Huang 2018 [36]	CECT	LightSpeed VCT	1.25 mm	-	-	-	-	-	Ultravist 370 1.5 mL/kg, 3.5 mL/s
Jeon 2021 [37]	mpMRI	1.5 or 3.0T	5 mm	4500/107	-	512 × 512	-	90	-
Jing 2022 [38]	mpMRI	GE Discovery 1.5 or 3.0T	4 mm	6538/116	20 × 20	352 × 352	32	110	Gd-DTPA 2 mL/s
Li 2022 [39]	mpMRI	Siemens Verio 3.0T	3 mm	-	26 × 26	202 × 288	-	10	Gd 0.1 mol/kg, 3.5 mL/s
Meng 2019 [40]	mpMRI	GE Optima 1.5	3 mm	4800–5000/102	-	288 × 256	24	-	Gd-DTPA 0.1 mmol/kg, 2 mL/s
Negreros-Osuna 2020 [41]	CECT	Multiple	5 mm	-	-	-	-	-	IsoVue 370 mg
Oh 2020 [29]	T2WI MRI	Philips Achieva 3.0T	3–5 mm	2500–8600/80–110	15 × 15–36 × 36	224 × 224	16–32	-	-
Zhang, G. 2021 [42]	mpMRI	GE Discovery 3.0T	3 mm	487–7355/8–136	20 × 20 to 34 × 34	128 × 140 to 352 × 256	-	-	-
Zhang, W. 2021 [43]	T2WI MRI	Siemens Magnetom Skyra 3.0T	3 mm	6890/100	18 × 18	384 × 346	-	-	-
Zhang, Z. 2021 [44]	T2WI MRI	GE Signa Horizon 3.0T	-	-	-	-	-	-	-

CECT: contrast-enhanced CT, mpMRI: multiparametric magnetic resonance imaging, SLT: slice thickness, TR: repetition time, TE: echo time, FOV: field of view, ACQ: acquisition, ETL: echo train length.

**Table 4 cancers-15-05816-t004:** Software and performance.

Study	Segmentation Software	Radiomics Software	Performance of Signature (Training)			Performance of Signature (Validation)			Outcome
			AUC	Sens	Spec	AUC	Sens	Spec	
Chen 2020 [34]	ITK-SNAP	Artificial Intelligence Kit v3.0	0.859	0.872	0.739	0.762	0.941	0.61	TOPO-iiα
Chen 2022 [28]	Syngo	Big Data Intelligent Analysis Cloud Platform	0.932	0.829	0.925	0.894	0.833	0.818	AQP-1
Horvat 2019 [35]	ITK-SNAP	-	-	-	-	-	-	-	APC, RASA1, ATM, BRCA2
Huang 2018 [36]	3D Slicer v4.3	MatLab 2013a	0.812	0.635	0.845	0.735	0.521	0.854	Tumour grade
Jeon 2021 [37]	3D Slicer v4.1	MatLab R2019b	0.76	-	-	0.729	-	-	CD8+ TIL
Jing 2022 [38]	Radcloud	Radcloud Radiomics Platform	0.910	0.844	0.929	0.874	0.909	0.815	MMR
Li 2022 [39]	Omni Kinetics	Omni Kinetics	0.853	0.9	0.733	0.747	0.882	0.615	LRP-1
			0.780	0.7	0.833	0.8	0.824	0.769	Survivin
Meng 2019 [40]	MITK v2013, ITK-SNAP	MatLab v2015a	0.707	0.713	0.607	0.696	0.667	0.574	HER-2
			0.607	0.622	0.536	0.699	0.863	0.422	Ki67
			0.752	0.774	0.612	0.677	0.762	0.494	Lymph nodes
			0.675	0.679	0.597	0.72	0.759	0.565	Differentiation
			0.669	0.531	0.748	0.651	0.581	0.643	KRAS
Negreros-Osuna 2020 [41]	TexRAD	TexRAD	-	-	-	-	-	-	BRAF
Oh 2020 [29]	-	MatLab	0.884	0.84	0.8				KRAS
Zhang, G. 2021 [42]	ITK-SNAP	Machine learning: 3D V-Net Radiomics: Pyradiomics	0.887	0.882	0.661				KRAS, NRAS, BRAF
Zhang, W. 2021 [43]	ITK-SNAP	Pyradiomics v2.1.2	0.989	-	-	0.895	0.667	0.987	MSI
Zhang, Z. 2021 [44]	MIM	MatLab	0.801	0.64	0.853	0.703	0.438	1	KRAS

TOPO: Topoisomerase, AQP: Aquaporin, APC: Adenomatous Polyposis Coli, RASA1: Rat Sarcoma A1, ATM: Ataxia-Telangiectasia Mutated, BRCA: Breast Cancer Gene, CD8+ TIL: Cluster of Differentiation 8 Tumour Infiltrating Lymphocyte, MMR: Mismatch Repair, LRP: Low-Density Lipoprotein Receptor-related Protein, HER: Human Epidermal Growth Factor Receptor, KRAS: Kirsten rat sarcoma, BRAF: B Rapidly Accelerated Fibrosarcoma, NRAS: Neuroblastoma rat sarcoma, MSI: Microsatellite Instability.

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
