# Peer review of "Radiogenomics: Contemporary Applications in the Management of Rectal Cancer"

_cancers, 2023, doi:10.3390/cancers15245816_

Round 1

Reviewer 1 Report

Comments and Suggestions for Authors

The enclosed manuscript intends to review the up-to-date performance of rectal cancer prognosis using imaging-based radiogenomic analysis. The review process followed the PRISMA guideline and provided sufficient information for the inclusive and exclusive criteria for literature searching. At last, 13 articles were included in the review for discussion. Basically, this is a well-organized and considerate draft, but some minor comments are as follows:

1) According to the description in the introduction, this article aims to demonstrate the strengths and weaknesses of the use of radiogenomics in predicting rectal cancer outcomes; however, it is unclear which particular field the authors would like to discuss in the texts. Whether the selection of the algorithm, the software, or the statistical analysis of the extractions remains unclear. 

2) It is not sure why the authors include the "impact factor" in table 1, which is full of speculation. Does it affect the credibility of the results or any weighting should we consider when a systemic review using those data? 

3) Two out of the 13 articles were CT scans. Despite feature extraction is an objective process, it is still a question why the authors don't focus in, for example, the MRI. In terms of the limited number of references, 13 is not too different from 11. 

4) Substantially, the authors claimed that this is an in-depth analysis of the current literature investigating the potential of replacing radiogenomics with genetic screening. I don't find those elements in the texts to convince readers that the current existing methods are good enough nor a clear suggestion on the computer model. The authors may want to add a sophisticated paragraph or segment to emphasize what the current progress indicates. 

Reviewer 2 Report

Comments and Suggestions for Authors

The article "Radiogenomics; Contemporary Applications in the Management of Rectal Cancer" by Niall J. O’Sullivan has several limitations.

Firstly, the study relies heavily on retrospective data, with 11 out of the 13 included studies being retrospective. This could introduce bias as retrospective studies are based on pre-existing data and may only account for some relevant factors.

Secondly, the article acknowledges considerable heterogeneity between study outcomes, which prevented the performance of a quantitative analysis]. This heterogeneity could limit the generalizability of the findings, as the results may only apply to some populations or settings.

Thirdly, the article excluded several studies for reasons such as incorrect study design, incorrect population, failure to develop radiogenic-based signatures, and lack of measurable outcomes. This could potentially limit the comprehensiveness of the review.

Fourthly, PRISM guidelines, QUADAS analysis, and publication bias were not considered.

In addition to these specific limitations, the field of radiogenomics itself faces several challenges. These include issues of standardization, as imaging acquisition, segmentation methods, and reconstruction protocols may differ. The repeatability and reproducibility of radiogenomics models is another significant limitation. Furthermore, the generalizability of results can be limited due to inter- and intra-institutional heterogeneity of datasets due to different hardware and scan protocols. Lastly, a lack of external validation is a pronounced limitation for the genetic validation of radiogenomic.

In conclusion, while the article provides valuable insights into the applications of radiogenomics in managing rectal cancer, these limitations should be considered when interpreting the findings. Future research should address these limitations to enhance the validity and applicability of the results.

Comments on the Quality of English Language

minor

Round 2

Reviewer 1 Report

Comments and Suggestions for Authors

The authors have addressed my concerns properly. 

Reviewer 2 Report

Comments and Suggestions for Authors

I am satisfied with the revisions.